# Adjuvant Chinese Medicine for the Treatment of Type 2 Diabetes Mellitus Combined with Mild Cognitive Impairment: A Systematic Review and Meta-Analysis of a Randomised Controlled Trial

**DOI:** 10.3390/ph15111424

**Published:** 2022-11-17

**Authors:** Changxing Liu, Xinyi Guo

**Affiliations:** Department of Integrative Medicine of the First Clinical Medical College, Shaanxi University of Traditional Chinese Medicine, Xianyang 712046, China

**Keywords:** diabetes mellitus, cognitive impairment, Chinese herbal medicine, systematic review, meta-analysis

## Abstract

Mild cognitive impairment has a high prevalence in the type 2 diabetic population. Adjuvant therapy with Chinese herbal medicine can effectively improve the clinical symptoms of patients with T2DM combined with MCI. The aim of this study was to systematically evaluate the efficacy and safety of Chinese herbal adjunctive therapy in the treatment of diabetes mellitus combined with cognitive impairment. Information was analysed using the China Knowledge Network, Vip Database, Wanfang Database, China Biomedical Literature Database, PubMed, EMbase, Web of Science, and MedLine Database. The total clinical efficiency, blood glucose, blood lipids, Simple Mental-State Examination Scale (MMSE), Montreal Cognitive Assessment Scale (MoCA), Traditional Chinese Medicine Symptom Score (TCMSS), and incidence of adverse reactions were recorded. The methodological quality of the included studies was evaluated using the application of the Cochrane Collaboration Network Risk Bias Assessment Tool, and meta-analysis was performed using RevMan 5.4 software. Adjuvant treatment with Chinese herbal medicine was effective in improving the clinical outcomes (OR = 5.33, 95% CI (3.62, 7.84), *p* < 0.00001) and cognitive function by comparing with the control group: MMSE (MD = 1.56, 95% CI (1.29, 1.84), *p* < 0.00001) and MoCA (MD = 2.77, 95% CI (1.81, 3.73), *p* < 0.0001); lowered blood glucose: fasting blood glucose (FBG) (MD = −0.27, 95% CI (−0.42, −0.12), *p* = 0.0006), 2 hPG (MD = −0.28, 95% CI (−0.45, −0.10), *p* = 0.002), and glycated haemoglobin (HbA1c) (MD = −0.26, 95% CI (−0.39, −0.14), *p* < 0.001); and improved lipids: total cholesterol (TC) (MD = −0.51, 95% CI (−0.82, −0.21), *p* = 0.001), triglycerides (TGs) (MD = −0.46, 95% CI −0.46, 95% CI (−0.80, −0.11), *p* = 0.009), low-density lipoprotein (LDL-C) (MD = −0.28, 95% CI (−0.55, −0.02), *p* = 0.04), high-density lipoprotein (HDL-C) (MD = 0.17, 95% CI (0.07, 0.28), *p* = 0.001), reduced TCMSS (MD = −1.84, 95% CI (−2.58, −1.10), *p* < 0.0001), and incidence of adverse events (OR = 0.46, 95% CI (0.24, 0.88), *p* = 0.02). In conclusion, through the available evidence, herbal adjuvant therapy for T2DM combined with MCI was observed to be effective and did not significantly increase the adverse effects. Due to the limitation of the number and quality of the included studies, the abovementioned results need to be validated by further high-quality studies.

## 1. Introduction

Type 2 diabetes mellitus (T2DM) combined with mild cognitive impairment (MCI) is characterised by memory loss as the primary symptom, involving attention and executive functions, often occurring in the complication phase of diabetes [1]. As the average age of the population continues to increase, at present, increasingly more studies are discovering that diabetes and underlying cognitive decline in the body share many standard features in the neuropathological process [2]. The studies have confirmed that diabetes can affect the brain’s structure and thus cause changes in its function, suggesting that people with diabetes, especially older people, are at a higher risk of cognitive decline and even dementia [3]. The epidemiological data show that people with diabetes have a 60% chance of developing MCI, which places a considerable emotional and financial burden on patients and their families [4]. Therefore, how to effectively intervene at the early stage of disease development to prevent the transformation and further development of the disease is gradually attracting attention from medical practitioners, the diabetic population, and their families. There are no ideal drugs or therapies to delay or improve the condition of patients with T2DM combined with MCI [5] that have been developed by modern medicine, and many researchers have turned their attention to complementary and alternative forms of medicine to discover some effective treatments.

In traditional Chinese medicine, T2DM combined with MCI is classified as “thirst” or “dullness”, which is caused by the interaction of multiple factors, such as old age and kidney deficiency, emotional and mental disorders, poor diet, and the depletion of a long-term illness. A kidney deficiency characterises pathogenesis as the primary cause, and phlegm, blood stasis, and qi stagnation as the symptoms [6,7,8]. The development of TCM is considered to be a dynamic evolutionary process. The disease originates as a result of prolonged thirst, the depletion of qi and yin, and the loss of yin and yang, resulting in the deficiency in both yin and yang and damage to all organs [9]. In recent years, clinical studies on this disease have achieved good efficacy in TCM, but the studies published in the literature are small, single-centre clinical trials and lack high-quality meta-summaries. As a result of this, the present study adopts an evidence-based approach to systematically evaluate the efficacy and safety of Chinese medicine in the adjuvant treatment of T2DM combined with MCI to provide an objective, evidence-based basis for clinical treatment and guided drug use.

## 2. Methods

### 2.1. Research Subjects

The diagnosis of T2DM was performed according to the “Chinese Guidelines for the Prevention and Treatment of Type 2 Diabetes Mellitus (2010 edition)” [10]. The diagnostic criteria for MCI were also based on the revised diagnostic criteria for AD-related MCI of the Alzheimer’s Association [11], specifically (1) no diagnosis of dementia; (2) intact life skills; (3) impaired objective memory; (4) generally normal cognitive function; (5) memory impairment; (6) a total score of <26 on the Montreal Cognitive Assessment Scale (MoCA) and a score of <26 on the Clinical Dementia Rating Scale (CDR); and a Clinical Dementia Rating (CDR) score of 0.5. 

### 2.2. Inclusion Criteria

(1) Study type: an RCT trial, whether blinded or not, with the language limited to Chinese and English; (2) interventions: the control group was treated with cognitive improvement drugs in addition to conventional diabetes treatment, and the trial group was treated with a Chinese herbal medicine compound in addition to the control group; and (3) outcome indicators: at least one of the indicators, including total clinical efficiency, blood glucose, blood lipids, MMSE, MoCA, and TCM symptom score (the TCM symptom score is a score of the patient’s TCM symptoms prior to and following treatment, according to the Diagnostic and Efficacy Criteria for TCM Evidence [12], Higher scores corresponded to more severe symptoms) and the incidence of adverse effects. (4) Good balance and comparability between groups.

### 2.3. Exclusion Criteria

(1) Duplicate publications; (2) literature with incomplete data or access to key information; and (3) reviews and conference abstracts.

### 2.4. Literature Search

A comprehensive search of the databases, such as China Knowledge Network, Vip Database, Wanfang Database, China Biomedical Literature Database, PubMed, EMbase, Web of Science, the Cochrane Library, and MedLine, was conducted until September 2022. Chinese search terms included “diabetes”, “diabetes cognitive impairment”, “diabetes cognitive impairment”, “Chinese medicine”, “Chinese herbal decoction”, and “proprietary Chinese medicine”, and the English search terms included “Diabetes, Diabetes cognitive impairment”, “Chinese Medicine”, and “Chinese Herbal Compound”, all of which were searched using subject terms + free words.

### 2.5. Data Extraction

A data extraction form was created, with two trained researchers extracting the data and adding another researcher to discuss and resolve any differences of opinion. The original metrics of the relevant literature were verified and validated, and the original authors could be contacted by email if there were errors or ambiguous information evident. If valid raw data could not be obtained, the problematic literature was discarded, and the quality of the original literature was strictly controlled before it was included in the meta-analysis.

### 2.6. Quality Evaluation

Two evaluators evaluated the extracted methodological features for the rigorous methodological quality of the included studies regarding the Cochrane Risk Bias Assessment Tool. If the literature was controversial, a third researcher screened and then read the full text and re-screened it. The data that ultimately met the criteria prerequisites were included in the analysis mapping the literature search and screening flow.

### 2.7. Statistical Methods

The meta-analysis was performed using the RevMan 5.3 software provided by the Cochrane Collaboration Network. Discontinuous variables were expressed as OR (odds ratio) and the continuous variables as MD (mean difference), with each effect size expressed as a 95% confidence interval (CI). The value of I2 < 50% indicated the lack of heterogeneity across the studies, and a fixed-effects model was used; conversely, statistical heterogeneity was indicated, and subgroup analyses were performed to eliminate heterogeneity according to possible heterogeneity factors. If statistical heterogeneity still existed, but clinical homogeneity was present, a meta-analysis was performed using a random effects model. Descriptive analyses were used if the heterogeneity was too great or the combination was deemed clinically inappropriate. When the number of papers combining the outcome indicators was ≥10, publication bias was analysed using funnel plots. The differences were considered statistically significant at *p* < 0.05.

## 3. Results

### 3.1. Literature Search Results and Basic Features

A total of 657 relevant papers were searched. Following the screening, 16 RCTs [13,14,15,16,17,18,19,20,21,22,23,24,25,26,27,28] with a total of 1299 patients were finally included in the study, of which all patients had essentially the same baseline level (see Figure 1 for the literature screening process and Table 1 for the literature characteristics).

### 3.2. Risk of Bias Results

The quality of the included literature was assessed using the Risk Assessment Tool recommended by the Cochrane Collaboration: 12 [13,14,15,16,17,18,19,21,23,24,26,27,28] mentioned the random-number table method; the remainder mentioned randomisation only, and were rated as “low risk”; the rest of the literature only mentioned randomisation and rated it as “unclear risk”; only one study mentioned allocation concealment [13]; and three mentioned a single-blind study [13,14,15] and rated it as “low risk”. The remaining studies did not mention allocation concealment or blinding, and were rated as “unclear risk”; all the literature presented clear outcome indicators and were rated as “low risk”; none of the studies obtained duplicate publications or publication bias and were evaluated as “low risk”; other biases were unknown and were evaluated as “low risk”; and all the data reported in the literature were complete and comparable between groups (Figure 2 and Figure 3).

### 3.3. Meta-Analysis Results

#### 3.3.1. Clinical Effectiveness

A total of 750 patients presented in 10 papers [13,14,18,20,21,22,23,24,25,26] were evaluated using clinical efficacy as an index. The results present homogeneity among the 10 included papers, and were therefore analysed using a fixed-effects model (I2 = 0%, [OR = 5.33, 95% CI (3.62, 7.84), *p* < 0.00001]). This suggests that the clinical efficacy of herbal adjuvant therapy for T2DM combined with MCI is superior to that of conventional western drugs, as shown in Figure 4. This indicates that adjuvant treatment using Chinese herbal medicine can significantly improve overall clinical efficiency compared to treatment using Western medicine alone.

#### 3.3.2. MoCA Scores

A total of 833 patients presented in 10 papers [13,14,15,18,19,20,21,24,27,28] were evaluated using the MoCA scale as an index. The results present heterogeneity in the 10 papers included in the study; therefore, the analysis was conducted using a random-effects model (I2 = 84%, [MD = 2.77, 95% CI (1.81, 3.73), *p* < 0.00001]). This suggests that adjuvant Chinese medicine improves the MoCA scores for T2DM combined with MCI more effectively than conventional Western medicines (see Figure 5).

#### 3.3.3. MMSE Scores

A total of 642 patients included in 10 papers [13,14,16,20,21,22,23,24,25,26] were evaluated using the MMSE scale as an index. The results present heterogeneity in the 10 papers included in the study; therefore, the analysis was presented using a random-effects model (I2 = 57%, [MD = 1.56, 95% CI (1.29, 1.84), *p* < 0.00001]). This suggests that adjuvant Chinese medicine improves the MMSE scores for T2DM combined with MCI more effectively than conventional Western medicine (see Figure 6).

#### 3.3.4. Chinese Medicine Symptom Score

A total of 320 patients included in 5 publications [13,17,20,26,28] used the TCM symptom score as an evaluation index. The results present homogeneity among the 4 included publications and were therefore analysed using a fixed-effects model (I2 = 25%, [MD = −1.84, 95% CI (−2.58, −1.10), *p* < 0.0001]). This suggests that adjuvant Chinese medicine improves the TCM symptom score of T2DM combined with MCI more effectively than conventional Western medicine (see Figure 7).

#### 3.3.5. Blood Glucose Indicators

Six studies [13,15,16,17,19,21] mentioned FBG as well as HbA1c, and three studies [13,16,19] mentioned 2hPG, which were analysed using a fixed-effects model by heterogeneity tests presenting homogeneity across the studies: FBG (MD = −0.27, 95% CI (−0.42, −0.12), *p* = 0.0006), 2 hPG (MD = −0.28,95% CI (−0.45, −0.10), *p* = 0.002), and HbA1c (MD = −0.26, 95% CI (−0.39, −0.14), *p* < 0.001). The results show that the test group presents lowered blood glucose levels at a better rate than the control group (see Figure 8).

#### 3.3.6. Blood Lipid Indicators

Two studies [15,18] mentioned TC, TG, LDL-C, and HDL-C results and other indicators, and presented homogeneity across the studies via a heterogeneity test; therefore, a fixed-effect model was used to perform the analysis: TC (MD = −0.51, 95% CI (−0.82, −0.21), *p* = 0.001), TG (MD = −0.46, 95% CI (−0.80, −0.11), *p* = 0.009), LDL-C (MD = −0.28, 95% CI (−0.55, −0.02), *p* = 0.04), and HDL-C (MD = 0.17, 95% CI (0.07, 0.28), *p* = 0.001). The results show that the level of lipid regulation in the test group is better than that in the control group (see Figure 9).

#### 3.3.7. Adverse Reactions

Six papers [13,15,19,20,21,28] reported adverse reactions, and the results present homogeneity in the six papers; therefore, the analysis used a fixed-effects model (I2 = 19%, [OR = 0.46, 95% CI (0.24, 0.88), *p* = 0.02]). This suggests that adjuvant Chinese medicine is less effective in improving the incidence of adverse effects in T2DM combined with MCI than conventional Western medicine (see Figure 10).

### 3.4. Publication Bias

The MMSE (Figure 11a) and MoCA (Figure 11b) scales present significant asymmetry in the funnel plots, indicating the presence of publication bias in the included studies, possibly due to the small number of included studies and the difficulty of including negative results. Other outcome indicators were challenging to plot due to the insufficient number of studies presenting insufficient test validity scores.

## 4. Discussion

According to the World Health Organisation (WHO), the prevalence of diabetes worldwide has been on the rise in the last 50 years. It is expected that by 2030, the number of people with diabetes worldwide will reach 592 million, and diabetes and its complications will become the seventh leading cause of death in the world [29]. This is why it is very important to actively and effectively control the development of diabetes. Iminosugars are a group of sugar analogues in which the oxygen atom of the sugar ring has been replaced by nitrogen. Due to their structural similarity to sugar, these compounds exhibit strong glycosidase inhibitory activity and can regulate the biosynthesis and hydrolysis of glycoproteins. Therefore, these inhibitors are expected to become therapeutic agents for diseases related to disorders of glucose metabolism [30,31,32,33].

IIn recent years, cognitive impairment associated with diabetes has been an increasing concern. It is associated with cognitive impairment. A meta-analysis of 30,000 patients (of whom approximately 16% had type 2 diabetes) and 11 studies presented an RR of 1.5 [95% CI (1.31, 1.74)] for dementia in people with diabetes [34]. Upon imaging, it can be observed that the primary manifestation of type 2 diabetes-related brain atrophy is a reduction in the volume of whole or localised brain tissue, with more pronounced cortical atrophy occurring in patients with type 2 diabetes than in those with Alzheimer’s disease alone [35]. At present, there are no specific clinical treatment options for patients with diabetic cognitive impairment primarily treated in the same way as patients with simple cognitive impairment or dementia. There are no specific treatments available [36]. Therefore, the search for Chinese and Western medicine methods for preventing and treating diabetic cognitive impairment has become a research hotspot, at present.

According to traditional Chinese medicine, the basic pathogenesis of thirst is yin deficiency and dry heat. If the disease is prolonged, the evidence of yin deficiency is presented by the lungs, spleen, and kidneys, which depletes kidney essence, and kidney essence is deficient in filling the marrow, resulting in insomnia and forgetfulness; or if kidney yin is deficient, the internal heat of yin deficiency burns the kidney ligaments, and the flow of qi and blood is disrupted, resulting in the congestion of brain ligaments and empty brain veins, resulting in dementia [37].

This study was guided by the theory of Chinese and Western medicine, from the perspective of the combination of Chinese and Western medicine, using evidence-based methods to confirm that the adjuvant treatment of patients with T2DM combined with MCI via Chinese medicine can increase overall clinical efficiency, improve cognitive function, and reduce the Chinese medicine symptom score and the occurrence of adverse effects, thus controlling the progression of the disease. An increasing number of studies have shown that TCM can not only alleviate the hyperglycaemic symptoms of patients with T2DM combined with MCI, but also improve cognitive function through the positive effects of reducing oxidative stress, protecting neural cell architecture, promoting vascular renewal, and regulating cerebral microcirculation [38]. Numerous scholars have used a combination of TCM and Western medicine to create therapeutic treatments, while simultaneously reducing Western medicine’s toxic side-effects and drug resistance to a certain extent. A study conducted by Zhang, Y [39] determined that Epimedium may improve brain glucose metabolism by increasing glucose transport and promoting glucose catabolism, and suggested that this improves the learning memory function in mice through the APP/PS1/Tau triple-transgenic AD (3 × Tg-AD) model. Rhodiola Rosea is effective in tonifying Qi and nourishing the blood, invigorating blood circulation, and resolving blood stasis. A study conducted by Yang, N [40] observed that Rhodiola Rosea glycosides could effectively improve learning memory ability, increase serum and kidney tissue superoxide dismutase (SOD) levels, and decrease serum and kidney tissue malondialdehyde (MDA) levels in rats with diabetes. The herb has a warming effect on the kidneys and is also considered to be an aphrodisiac. Mao, X.Y et al. [41] concluded that the neuroprotective effect of Cnidii frutus on rats with diabetic encephalopathy might be mediated by inhibiting the PI3K/Akt signalling pathway to reduce the inflammatory response. Polygonum multiflorum belongs to the heart, liver, and kidney meridians, and has the effect of tonifying the liver and kidneys, and nourishing the heart and blood. In a study conducted by Tang, J [42], polygonum multiflorum was observed to downregulate the expression of MLCK and NR2B and upregulate the expression of the Wnt/β-catenin pathway in the hippocampus of diabetic rats, thereby improving the structure of hippocampal neurons and enhancing learning and memory capacities, which in turn improved diabetic cognitive dysfunction. Huang, X.B et al. [43] observed that combining Acorus calamus and Polygala could reduce serum methylglyoxal (MG) levels and improve cognitive impairment in diabetic rats. In addition, modern pharmacological studies have shown that Thatrhizoma Ligustici can improve cerebral blood circulation and reduce ischaemic damage and neurological dysfunction occurring in brain tissues. Its active ingredient, Thatrhizoma Ligustici, has the effect of upregulating the expression of brain-derived neurotrophic factors in the hippocampus [44], which improves cognition. Astragalus extract increased Bcl-2 and Bcl-xl expressions. It inhibited apoptosis in brain cells, improving cognitive learning and memory in rats, a model of cognitive impairment for diabetes [45]. Lycium barbarum seed oil obtained from Lycium barbarum reduced MCI in rats by inhibiting hippocampal acetylcholinesterase activity (TCh E) and increasing acetylcholine transferase (ChAT) content [46]. A study determined that Eucommia polysaccharides significantly increased SOD and glutathione peroxidase (GSH-Px) levels and decreased MDA levels in the serum of mice with diabetes mellitus, which may be one of the mechanisms of Eucommia treatment for diabetes mellitus [47]. Most kidney tonic herbs have a preventive and curative effect on the atrophy of hippocampal volume in MCI patients [48].

In addition, this study also included blood glucose and lipid indexes for analysis. The effect of changes in glucose and lipid metabolism on cognitive dysfunction in diabetes is more intuitive. Liu, Z.H et al. [49] conducted a correlation analysis of cognitive function scores and HbA1c levels in T2DM. They observed that long-term memory, associative learning and comprehension memory in short-term memory, and total memory scores on the Wechsler Memory Scale were negatively correlated with HbA1c levels. Cao, Z. et al. [50] observed that the higher the TC and HDL-C values, the better the cognitive function; the higher the TG and LDL-C values, the worse the cognitive function. The present study demonstrated that the adjuvant treatment of T2DM combined with MCI with Chinese herbal medicine could effectively lower blood glucose levels and regulate lipid metabolism, thus improving blood flow and protecting neurons.

The shortcomings of this study included: (1) a lack of detailed descriptions of the design methods and the concealment of randomised protocols in the 16 included studies, a lack of multicentre and blinded applications, and a possibility of implementation and measurement biases; (2) the funnel plot showed that publication bias was difficult to avoid because negative results often are difficult to report; and (3) a lack of follow-up sessions in the included studies and lack of presentation of the long-term effects and safety evaluations of Chinese medicine.

## 5. Conclusions

Based on the evidence presented in this study, for the treatment of T2DM combined with MCI, adjuvant Chinese medicine can effectively increase the overall clinical efficiency, improve cognitive function, regulate blood glucose and lipid levels, and reduce the TCM symptom score and incidence of adverse effects significantly better than the group using Western medicine alone, and presents a higher safety profile, providing objective evidence for the complementary treatment of this disease. However, due to various limitations, the results of this study still need to be validated by future higher-level clinical studies and primary trials.

## Figures and Tables

**Figure 1 pharmaceuticals-15-01424-f001:**
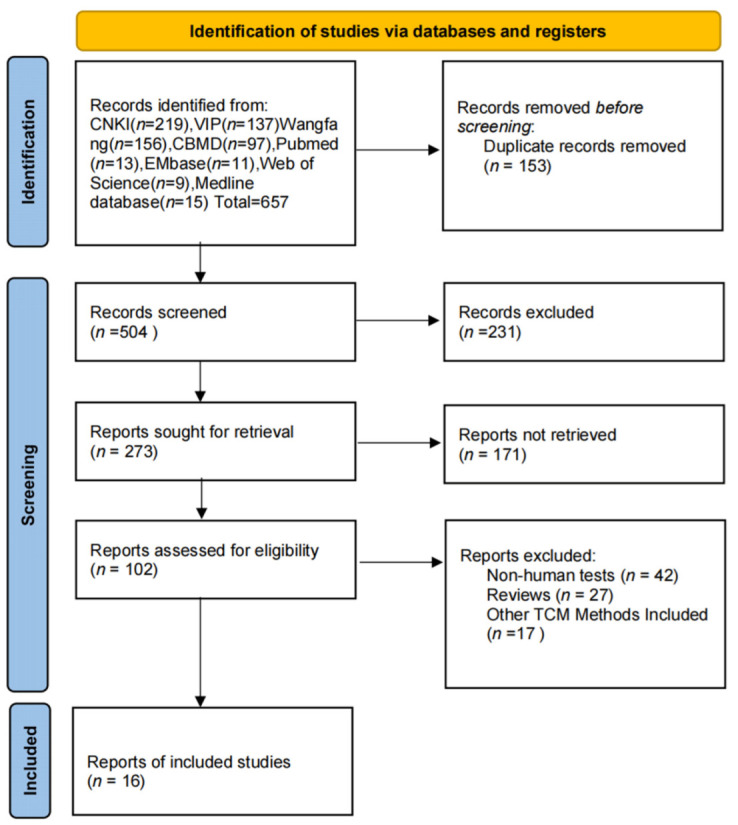
Flow chart of literature screening.

**Figure 2 pharmaceuticals-15-01424-f002:**
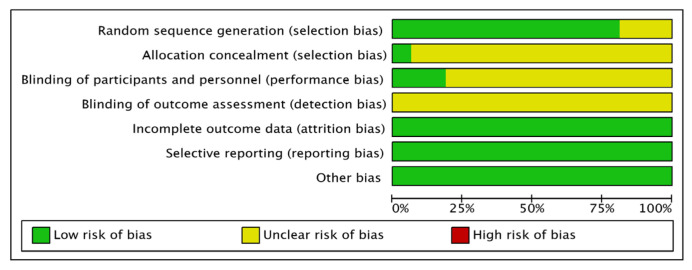
Summary of the risk of bias in the included literature.

**Figure 3 pharmaceuticals-15-01424-f003:**
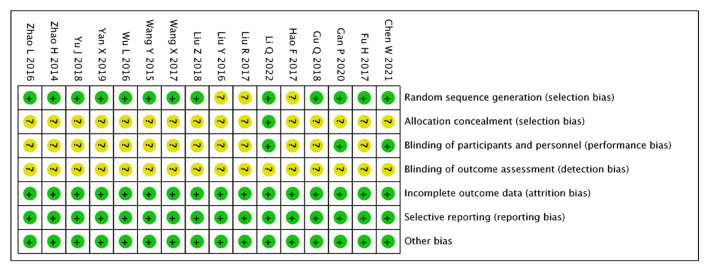
Risk of bias proportional to the risk of inclusion in the literature.

**Figure 4 pharmaceuticals-15-01424-f004:**
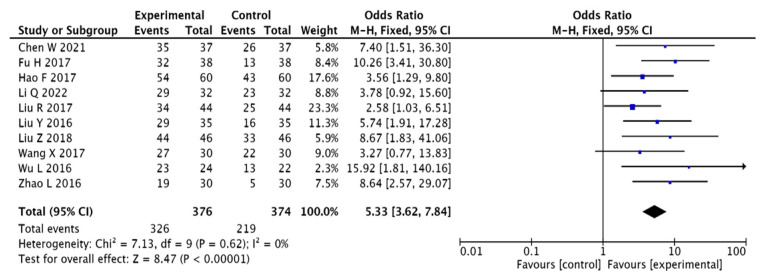
Forest diagram with efficiency comparisons [13,14,18,20,21,22,23,24,25,26].

**Figure 5 pharmaceuticals-15-01424-f005:**
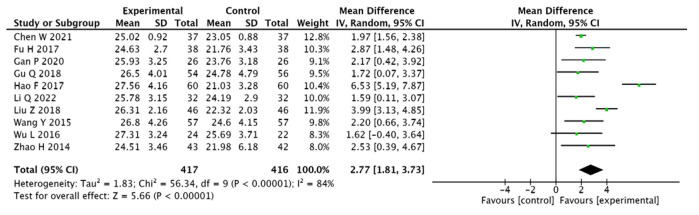
Forest plot of MoCA scores [13,14,15,18,19,20,21,24,27,28].

**Figure 6 pharmaceuticals-15-01424-f006:**
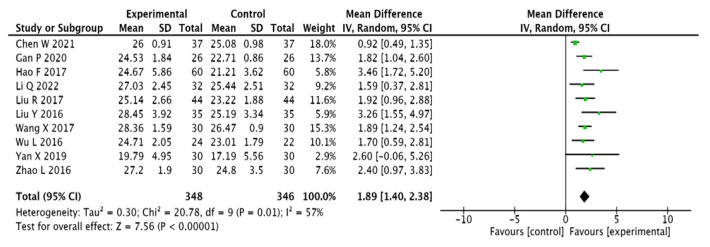
Forest plot of MMSE scores [13,14,16,20,21,22,23,24,25,26].

**Figure 7 pharmaceuticals-15-01424-f007:**
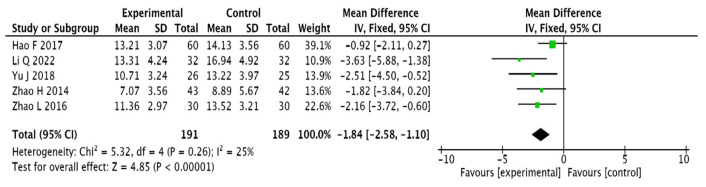
Forest plot of TCM symptom score [13,17,20,26,28].

**Figure 8 pharmaceuticals-15-01424-f008:**
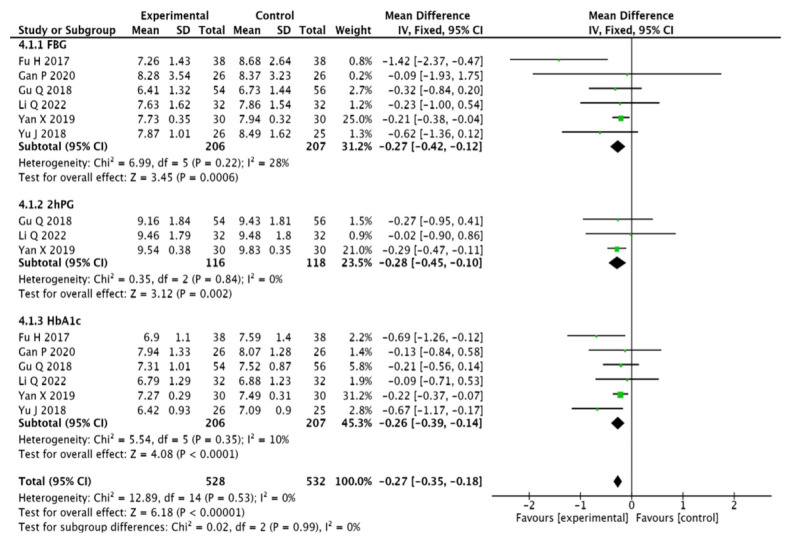
Meta-analysis forest plot of blood glucose [13,15,16,17,19,21].

**Figure 9 pharmaceuticals-15-01424-f009:**
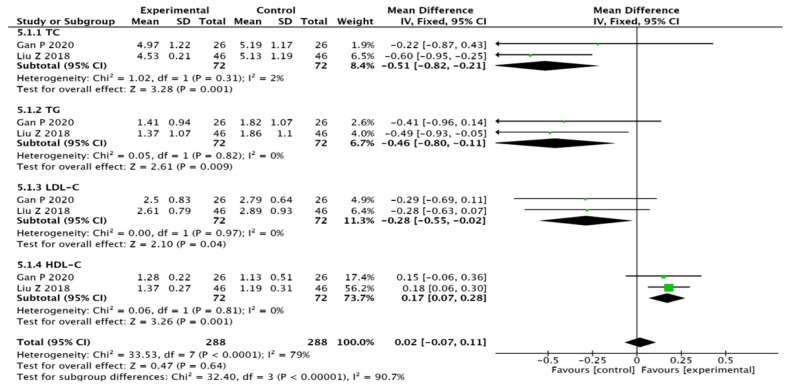
Meta-analysis forest plot of blood lipids [15,18].

**Figure 10 pharmaceuticals-15-01424-f010:**
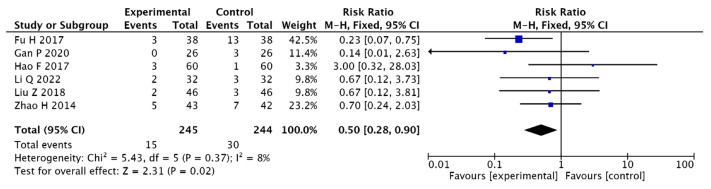
Forest plot of adverse reactions [13,15,19,20,21,28].

**Figure 11 pharmaceuticals-15-01424-f011:**
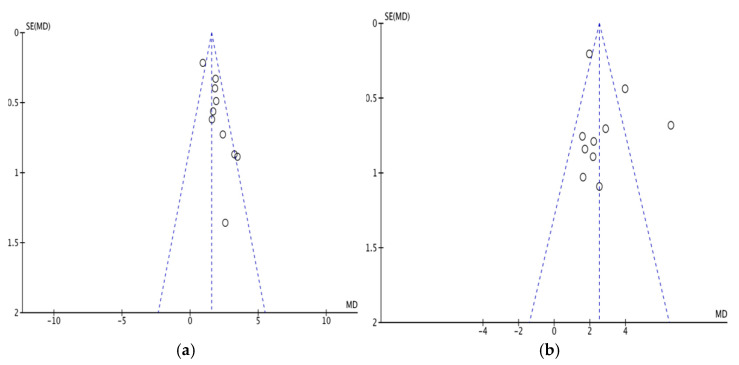
(**a**) MMSE bias funnel plot. (**b**) MoCA bias funnel plot.

**Table 1 pharmaceuticals-15-01424-t001:** Basic characteristics of the included literature.

Author	Case (T/C)	Intervening Measure		Time (W)	Out Come	Random Method	Adverse Reaction
		T	C				
Li Quan 2022 [13]	32/32	Dihuang Yinzi Soup +C	Donepezil tablets	12	①,②,③,④,⑥,⑦	Random number table	T:2 cases of dizziness; C: 3 cases of Tinnitus occurred
Chen Weiming 2021 [14]	37/37	Naolin soup+C	Donepezil tablets	24	①,②,③,④	Random number table	None
Gan Panpan 2020 [15]	26/26	Bushen Qingnao Pobi soup+C	Donepezil tablets	12	②,④,⑤,⑦	Random number table	C:2 cases of diarrhea; 1 case of headache
Yan Xiaoyan 2019 [16]	30/30	Yizhi granule+C	Nimodipine	24	②,③	Only described as random	None
Yu Jinxin 2018 [17]	26/25	Xuefu Zhuyu Soup+C	Donepezil tablets	8	②,⑥	Random number table	None
Liu Zhonghua 2018 [18]	46/46	Yiben huoxue recipe+C	Nimodipine	8	①,②,⑤,⑦	Random number table	T:5 cases of constipation; C:3 cases of skin rash
Gu Qunshan 2018 [19]	54/56	Bushen Yinao granule+C	Donepezil tablets	24	②,③,④	Random number table	None
Hao Fengqing 2017 [20]	60/60	Zishen Huatan Qushi soup+C	Donepezil tablets	24	①,③,④,⑥,⑦	Only described as random	C:2 cases of dizziness
Fu Hong 2017 [21]	38/38	Bushen Huoxue granule+C	Nimodipine	12	①,②,④,⑦	Random number table	T:3 cases of constipation; C:13 cases of dizziness
Liu Runping 2017 [22]	44/44	Bushen Quyu Yizhi Soup+C	Nimodipine	8	①,③	Only described as random	None
Wang Xiaoyan 2017 [23]	30/30	Xingnao Yizhi soup+C	Nimodipine	12	①,③	Random number table	None
Wu Lijuan 2016 [24]	24/22	BuShen Huoxue Kaiqiao recipe+C	Nicergolin tablets	24	①,③,④	Random number table	None
Liu Yafen 2016 [25]	35/35	Yiqi Bushen Huoxue recipe+C	Nicergolin tablets	12	①,③	Only described as random	None
Zhao Lihua 2016 [26]	30/30	Buxu Quzhuo Tongluo soup+C	Donepezil tablets	12	①,③,⑥	Random number table	None
Wang Yu 2015 [27]	57/57	Bushen Huoxue soup+C	Nimodipine	12	②,③	Random number table	None
Zhao Huan 2014 [28]	43/42	Bushen Huoxue Kaiqiao granule+C	Nimodipine	12	④,⑥,⑦	Random number table	T:7 cases of dizziness C:7 cases of nausea and vomiting

① Effective rate; ② Blood glucose; ③ MMSE; ④ MoCA; ⑤ Blood lipids; ⑥ TCMSS; ⑦. Adverse reactions; T:Treatment group; C:Control group.

## Data Availability

The datasets used during the study are available from the corresponding author upon reasonable request.

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
