# Peer review of "Adjuvant Chinese Medicine for the Treatment of Type 2 Diabetes Mellitus Combined with Mild Cognitive Impairment: A Systematic Review and Meta-Analysis of a Randomised Controlled Trial"

_pharmaceuticals, 2022, doi:10.3390/ph15111424_

Round 1

Reviewer 1 Report

In the article ‘Adjuvant Chinese medicine for the treatment of type 2 diabetes mellitus combined with mild cognitive impairment: a systematic review and meta-analysis of a randomised controlled trial, the author tries to review the traditional Chinese medicine and type 2 diabetes

mellitus with mild cognitive impairment. It is a valuable review.  However, there are some shortcomings and questions.

1)     In the line 9-13: Randomized controlled trials on the treatment of diabetes mellitus combined with cognitive impairment were searched by computer from China Knowledge Network, Wipe database, Wanfang database, China Biomedical Literature Database, PubMed, Embase, Web of Science, the Cochrane Library, MedLine database. Randomized or randomized? It is inconsistent in the whole paper.

2)     In the line 26-27:  …reduced TCM symptom score (MD=-1.84,95%CI(-2.58,1.10),P<0.0001) and incidence of adverse event (RR=0.50,95%CI(0.28,0.90),P=0.02).

Since the TCM is firstly presented in this paper, please provide the full name for TCM, and introduce the TCM symptom score in the Introduction and Methods parts.

3)     The English language should be improved overall.

4)     All the figures are almost lacking figure legends. They are important for readers. Please provide them.

5)     In the line 129:  â‘ Effective rate;②Blood glucose;③ MMSE;④ MoCA;⑤ Blood lipids;⑥ TCMSS; ⑦.Adverse reactionsï¼›The format is inconsistent.

Author Response

Response to Reviewer 1

Dear reviewer:

Thank you for your decision and constructive comments on my manuscript. We have carefully considered the suggestion of Reviewer and make some changes. We have tried our best to improve and made some changes in the manuscript.

  1. In the line 9-13:Randomized controlled trials on the treatment of diabetes mellitus combined with cognitive impairment were searched by computer from China Knowledge Network, Wipe database, Wanfang database, China Biomedical Literature Database, PubMed, Embase, Web of Science, the Cochrane Library, MedLine database. Randomized or randomized? It is inconsistent in the whole paper.
  2. In the line 26-27:  …reduced TCM symptom score (MD=-1.84,95%CI(-2.58,1.10),P<0.0001) and incidence of adverse event (RR=0.50,95%CI(0.28,0.90),P=0.02).Since the TCM is firstly presented in this paper, please provide the full name for TCM, and introduce the TCM symptom score in the Introduction and Methods parts.
  3. The English language should be improved overall.
  4. All the figures are almost lacking figure legends. They are important for readers. Please provide them.
  5. In the line 129:  â‘ Effective rate;②Blood glucose;③ MMSE;④ MoCA;⑤ Blood lipids;⑥ TCMSS; â‘¦.Adverse reactionsï¼›The format is inconsistent.

Responses:

1)We thank the reviewers for their carefulness in finding the subtle problems in my paper. We have revised the randomized to Randomized in the full text.

2)Thank you for your comment, we have made changes as you suggested and introduced the TCM symptom score in the methodology.

3)Thank you for your comment, we have had the full text revised by a native English speaking professional.

4)Dear reviewers, after reviewing my manuscript, I found that the figures in the manuscript all have figure legends.

5)Revisions have been made based on your comments, thank you for your comments.

Reviewer 2 Report

Patients with Type 2 Diabetes Mellitus (T2DM) combined with Mild Cognitive Impairment (MCI) often had a dismal prognosis. The authors tried to evaluate the efficacy and safety of traditional Chinese medicine (TCM) as an adjuvant in treating T2DM combined with MCI by meta-analysis. This work is very interesting, nevertheless, I have few suggestions to make regarding the manuscript: 

ABSTRACT:

1. In Objective, the TCM seemed to be the major treatment, the word adjuvant is missing.

2. In Method, a database is Wipe database, while in Line 86, it was Weeipu database, are they the same?

3. In Results, the subject of the first sentence is missing. The significant digits in some results were 3 (e.g. P=0.46), while some were 6 (e.g. P<0.00001), please unify. The p and P were shown in different results.

INTRODUCTION:

1. What do you mean in the body? (Line 39) What do you mean “ families of origin”? (Line 44) Please ask an English native speaker to edit your expressions throughout the manuscript.

2. Line 48 to Line 50 “Modern medicine has no idea drugs or therapies to delay and improve the condition of patients with T2DM with MCI, and many researchers have turned their attention to complementary and alternative medicine to find some effective treatments.” Maybe the authors mean no ideal drugs. And please add some references to support this sentence.

METHODS

1. Why the diagnosis of T2DM is a Chinese one? In that case, you might fail to discover the studies outside China. Why did not use a more popular criteria, e.g. ICD-11 instead?

2. In 2.2, what do you mean “Chinese herbal medicine compound”? As I know, the Yizhi granule contains shark oil, while Yiben huoxue recipe contains Pheretima and powder of Hirudo nipponica Whitman, and they are not herb at all.

3. In 2.3, how about the retracted publications? The conference summary did not set as an exclusion criterion, but in Figure 1 some publications were excluded due to conference summary.

4. In 2.4, what do you mean "Chinese medicine Chinese medicine", "Chinese medicine" (Line 90)? They all look the same. Please provide a supplementary table of search terms in both Chinese and English for repetition and validation, but not use etc. instead.

5. In 2.5 and 2.6, who did the data extraction, quality evaluation, and made the decision respectively? Since you did not provide a Author Contribution, then please list their names here.

6. In 2.7, what do you mean OR (Line 110) and MD (Line 111). Please provide the full name when the abbreviation was first appeared.

RESULTS

1. In Figure 1, how much literature did you get from each database, please provide a exact number. The excluded literature should place to the right, but not the down arrow.

2. In Table 1, please provide the reference of each study. What do you mean - in Random Method? Not given?

DISCUSSION

The authors introduced lots of studies of TCM as an adjuvant for T2DM plus MCI (Line 247 to Line 291), however, they failed to explain their results. What are the components reported in those 16 RCTs respectively? Why those drugs were able to attenuate or alleviate T2DM plus MCI? Please explain it based on TCM theory and experimental evidence respectively. Besides, use the Latin name of each drugs, but not Chinese pronunciations.

REFERENCES

The reference style is not unified, even some of them were wrong (e.g. Reference 36, in Line 267, the authors name is Mao Xiaoyuan, but in Line 403, the authors name is Mao SY).

Author Response

Response to Reviewer 2

Dear reviewer:

Thank you for your decision and constructive comments on my manuscript. We have carefully considered the suggestion of Reviewer and make some changes. We have tried our best to improve and made some changes in the manuscript.

Response to Reviewer:

ABSTRACT:

  1. In Objective, the TCM seemed to be the major treatment, the word “adjuvant” is missing.

Answer:We thank the reviewers for identifying the problems, and we have revised them.

  1. In Method, a database is “Wipe database”, while in Line 86, it was “Weeipu database”, are they the same?

Answer:Many thanks to the reviewer for finding the problem, it was my mistake and we have revised it.

  1. In Results, the subject of the first sentence is missing. The significant digits in some results were 3 (e.g. P=0.46), while some were 6 (e.g. P<0.00001), please unify. The “p” and “P” were shown in different results.

Answer:Thank you for your comment, we have reorganised the abstract section in line with journal requirements.

INTRODUCTION:

  1. What do you mean “in the body”? (Line 39) What do you mean “ families of origin”? (Line 44) Please ask an English native speaker to edit your expressions throughout the manuscript.

Answer:Thank you for your comment, we have had the full text revised by a native English speaking professional.

  1. Line 48 to Line 50 “Modern medicine has no idea drugs or therapies to delay and improve the condition of patients with T2DM with MCI, and many researchers have turned their attention to complementary and alternative medicine to find some effective treatments.” Maybe the authors mean “no ideal drugs”. And please add some references to support this sentence.

Answer: I thank the reviewers for their comments; it was my mistake that the manuscript was incorrectly translated. There are currently no ideal drugs in modern medicine to delay and improve the condition of patients with T2DM with MCI, as according to the latest treatment guidelines for patients with T2DM combined with MCI, the pharmacological treatment of patients with T2DM combined with MCI is similar to that of non-diabetic patients and there are no recommended drugs with significant efficacy. For patients with a clear diagnosis of T2DM combined with MCI, cholinesterase inhibitors may be used; for patients with moderately severe disease, high-dose cholinesterase inhibitors may be used.

METHODS

  1. Why the diagnosis of T2DM is a Chinese one? In that case, you might fail to discover the studies outside China. Why did not use a more popular criteria, e.g. ICD-11 instead?

Answer:Thank you for your comment. After searching for subject terms and free words, the RCTs that were included were all studies done by scientists in mainland China, and the diagnostic criteria they used to include their subjects were in accordance with the "criteria in the Chinese guidelines for type 2 diabetes".

  1. In 2.2, what do you mean “Chinese herbal medicine compound”? As I know, the Yizhi granule contains shark oil, while Yiben huoxue recipe contains Pheretima and powder of Hirudo nipponica Whitman, and they are not “herb” at all.

Answer:Dear Reviewer, We found by checking with the State Drug Administration of China that the composition of puzzle granules includes: Gynostemma pentaphylla; pseudo-ginseng; ginseng; epimedium; Fructus Alpiniae Oxyphyllae; mulberry The composition does not contain shark oil. Pheretima and powder of Hirudo nipponica Whitman, both of which are herbal medicines, are contained in Yiben Revitalising Blood Soup. In Chinese herbal pharmacology, it is found that powdered drugs are more easily absorbed by the body when dissolved in water, which is why they are polished into powder in clinical practice.

  1. In 2.3, how about the retracted publications? The “conference summary” did not set as an exclusion criterion, but in Figure 1 some publications were excluded due to “conference summary”.

Answer: Thanks to the reviewer for the heads up and we have revised it.

  1. In 2.4, what do you mean "Chinese medicine Chinese medicine", "Chinese medicine" (Line 90)? They all look the same. Please provide a supplementary table of search terms in both Chinese and English for repetition and validation, but not use “etc.” instead.

Answer: Dear Reviewer, As I made an error in the English translation when writing the manuscript, thank you very much for pointing it out. Chinese search terms include "diabetes", "diabetes cognitive impairment", "diabetes cognitive impairment", "Chinese medicine ", "Chinese herbal decoction", "proprietary Chinese medicine", and the English search terms "Diabetes, Diabetes cognitive impairment ", "Chinese Medicine", "Chinese Herbal Compound", all of which were searched using subject terms + free words.

  1. In 2.5 and 2.6, who did the data extraction, quality evaluation, and made the decision respectively? Since you did not provide a “Author Contribution”, then please list their names here.

Answer: Author Contributions:Liu Changxing  conceived and designed the study, analysed the findings, wrote most of the article, including the abstract and body of the article, figures and tables, and conducted a literature search in databases. Xinyi Guo extracted and collated the data from the selected trials, and Xinyi Guo corrected and amended errors in the article and performed the statistical analysis and interpretation of the results. All authors have read and agreed to the published version of the manuscript.

  1. In 2.7, what do you mean “OR” (Line 110) and “MD” (Line 111). Please provide the full name when the abbreviation was first appeared.

Answer:Revisions have been made in response to your comments.The OR is the ratio of the number of events in the test group to the ratio of the number of events in the control group.The MD is the difference between the mean value of the test group and the mean value of the control group (MD is the difference of the means)

RESULTS

  1. In Figure 1, how much literature did you get from each database, please provide a exact number. The excluded literature should place to the right, but not the down arrow.

Answer:Thank you for your comment, which we have amended.

  1. In Table 1, please provide the reference of each study. What do you mean “-” in “Random Method”? Not given?

Answer:A search of the included studies of RCTs revealed that the authors simply mentioned randomisation and did not describe specifically what methods were used to carry out the randomisation.

DISCUSSION

The authors introduced lots of studies of TCM as an adjuvant for T2DM plus MCI (Line 247 to Line 291), however, they failed to explain their results. What are the components reported in those 16 RCTs respectively? Why those drugs were able to attenuate or alleviate T2DM plus MCI? Please explain it based on TCM theory and experimental evidence respectively. Besides, use the Latin name of each drugs, but not Chinese pronunciations. 

Answer:Dear reviewers, the 16 RCTs we included contained a total of 127 herbal medicines, however, our study focused on 10 herbal medicines in the discussion section because they appeared more than 3 times in the included RCTs, which is why we focused on their mechanism of action. In addition, we have also renamed the names of the herbs.

REFERENCES

The reference style is not unified, even some of them were wrong (e.g. Reference 36, in Line 267, the author’s name is Mao Xiaoyuan, but in Line 403, the author’s name is Mao SY).

Answer:Thank you for your comment, which we have amended.

Reviewer 3 Report

In this review, the authors have analyzed the efficiencies of Traditional Chinese Medicine (TCM) for the treatment of type 2 diabetes 2 mellitus (T2DM) combined with mild cognitive impairment (MCI) using meta-analysis methods systematically. In some literatures they have found that TCM is more effective than traditional medicines. Usually, traditional medicines are growing much interest due to their less kind of side effects. So, this review seems to be highly significant for corresponding topic.

My comments are:

1. The title of the manuscript is written correctly and represents the objective of the review.

2. The abstract is not presented in line with the journal format. A brief discussion of the manuscript should be presented. (No need to mention objective, methods, results and conclusions)

3. The introduction section is up to the mark but, the author should elaborate the use of TCM Traditional Chinese Medicine thoroughly with proper schematic representation (for better understanding) and references.

4. The methodologies used to analysis the data in the literature is found to be rational. All the search engines, software used for data comparison is authentic.

a)  All the abbreviations should be decoded properly.

b) The table 1 (Basic characteristics of the included literature) should be prepared properly to understand clearly. Inclusion of reference number is recommended for all 16 entries. (T and C should be explained).

c) From figure 4-8, inclusion of references is required.

d) From results, it is observed that TCM reduced both T2DM and MCI in in-vivo analysis as comparted to western medicines. But, comparison data with western antidiabetic medication is not presented?

c) The combination study of TCM and anti-Alzheimer medicine (Nimodipine, Donepezil tablets) was performed. What is the rationale behind this?

d)  The study of safety data and side effects are not evaluated properly.

5. The author claimed that “Modern medicine has no idea drugs or therapies to delay and improve the condition of patients with T2DM with MCI.” To justify this statement, authors should provide sufficient data. (GLP 1 agonists are found to be good therapy for T2DM with MCI).

Please go for the article:

(Int. J. Mol. Sci. 2022, 23(9), 4641; https://doi.org/10.3390/ijms23094641)

6. All the references should be formatted properly in line with the journal guidelines.

7. It is recommended to emphasis the importance of iminosugars and sugar derivatives as an anti-diabetic agents, and it is suggested to cite following relevant articles related to iminosugars in introduction section.

  1. Nash, R. J.; Kato, A.; Yu, C-. Y.; Fleet, G. W. J. Iminosugars as therapeutic agents: recent advances and promising trends. Future Med. Chem. 2011, 3, 1513−1521.
  2. Yang, L.-F.; Shimadate, Y.; Kato, A.; Li, Y.-X.; Jia, Y.-M.; Fleet, G.W.J.; Yu, C.-Y. Synthesis and glycosidase inhibition of N-substituted derivatives of DIM. Org. Biomol. Chem. 2020, 18, 999–1011.
  3. Chennaiah, A.; Bhowmick, S.; Vankar, Y. D. Conversion of glycals into vicinal-1,2-diazides and 1,2-(or 2,1)-azidoacetates using hypervalent iodine reagents and Me3SiN3. Application in the synthesis of N-glycopeptides, pseudo-trisaccharides and an iminosugar. RSC Adv. 2017, 7, 41755−41762.
  4. Rajasekaran, P.; Ande, C.; Vankar, Y. D. Synthesis of (5,6 & 6,6)-oxa-oxa annulated sugars as glycosidase inhibitors from 2-formyl galactal using iodocyclization as a key step. ARKIVOC 2022, vi, 5−23.
  5.  

Overall, after addressing the points mentioned above, I recommend this review to publish in Pharmaceuticals.

Author Response

Response to Reviewer 3

Dear reviewer:

Thank you for your decision and constructive comments on my manuscript. We have carefully considered the suggestion of Reviewer and make some changes. We have tried our best to improve and made some changes in the manuscript.

Response to Reviewer:

  1. The title of the manuscript is written correctly and represents the objective of the review.

Answer:Thank you for your comments.

  1. The abstract is not presentedin line with the journal format. A brief discussion of the manuscript should be presented. (No need to mention objective, methods, results and conclusions)

Answer:Thank you for your comment, we have reorganised the abstract section in line with journal requirements.

  1. The introduction section is up to the mark but, the author should elaborate the use of TCM Traditional Chinese Medicine thoroughly with proper schematic representation (for better understanding) and references.

Answer:We thank the reviewers for their comments, which are valuable to us. This study has already described the understanding and treatment of patients with T2DM combined with MCI in Chinese medicine in the discussion section, as well as describing previous studies on the mechanism of action of Chinese medicine in the disease.

  1. The methodologies used to analysis the data in the literature is found to be rational. All the search engines, software used for data comparison is authentic. 
  2. a)  All the abbreviations should be decoded

Answer:Thank you for your comment, which we have amended.

  1. b) The table 1(Basic characteristics of the included literature) should be prepared properly to understand clearly. Inclusion of reference number is recommended for all 16 entries. (Tand Cshould be explained).

Answer:Thank you for your comment, which we have amended.

  1. c) From figure 4-8, inclusion of references is required. 

Answer:Thank you for your comment, which we have amended.

  1. d) From results, it is observed that TCM reduced both T2DM and MCI in in-vivo analysis as comparted to western medicines. But, comparison data with western antidiabeticmedication is not presented? 

Answer:We thank the reviewers for their comments. In the RCTs included in this study, both the test group and the control group were already taking conventional hypoglycaemic drugs, and the control group was treated with conventional diabetes treatment plus cognitive improvement drugs, while the test group was treated with a Chinese herbal medicine combination on top of the control group. In addition, the purpose of this study was to compare the efficacy of Chinese herbal medicine combined with conventional drugs for improving cognitive function with conventional drugs for improving cognitive function alone, so no comparison was made with conventional hypoglycaemic drugs.

  1. c) The combination study of TCMand anti-Alzheimer medicine(Nimodipine, Donepezil tablets) was performed. What is the rationale behind this?

Answer:We thank the reviewers for their valuable comments. According to the latest guidelines for the treatment of patients with T2DM combined with MCI, cholinesterase inhibitor therapy is available for patients with a clear diagnosis of T2DM combined with MCI, and high-dose cholinesterase inhibitor therapy is available for patients with moderate to severe disease.

Source: Chinese Society of Endocrinology, Chinese Working Group on Targeted Antihypertensive Therapy in Adults with Type 2 Diabetes. Chinese Journal of Endocrinology and Metabolism,2022,38(06):453-464.

  1. d)  The study of safety data and side effects are not evaluated properly.

Answer:Thank you for your comments, the adverse event has been re-rated.

  1. The author claimed that “Modern medicine has no idea drugs or therapies to delay and improve the condition of patients with T2DM with MCI.”To justify this statement, authors should provide sufficient data. (GLP 1 agonists are found to be good therapy for T2DM with MCI).

Answer:I thank the reviewers for their comments; it was my mistake that the manuscript was incorrectly translated. There are currently no ideal drugs in modern medicine to delay and improve the condition of patients with T2DM with MCI, as according to the latest treatment guidelines for patients with T2DM combined with MCI, the pharmacological treatment of patients with T2DM combined with MCI is similar to that of non-diabetic patients and there are no recommended drugs with significant efficacy. For patients with a clear diagnosis of T2DM combined with MCI, cholinesterase inhibitors may be used; for patients with moderately severe disease, high-dose cholinesterase inhibitors may be used.

Source: Chinese Society of Endocrinology, Chinese Working Group on Targeted Antihypertensive Therapy in Adults with Type 2 Diabetes. Chinese Journal of Endocrinology and Metabolism,2022,38(06):453-464.

  1. All the references should be formatted properly in line with thejournal guidelines.

Answer:Thank you for your comment, which we have amended.

  1. It is recommended to emphasis the importance of iminosugars and sugar derivatives as an anti-diabetic agents, and it is suggested to cite following relevant articles related to iminosugars in introduction section.

Answer:We thank the reviewers for their comments. After reading the literature, we believe that iminosaccharides and sugar derivatives are indeed important in diabetes, so we decided to cite him in the discussion section.